# Coefficient Extraction of SAC305 Solder Constitutive Equations Using Equation-Informed Neural Networks

**DOI:** 10.3390/ma16144922

**Published:** 2023-07-10

**Authors:** Cadmus Yuan, Qinghua Su, Kuo-Ning Chiang

**Affiliations:** 1Department of Mechanical and Computer-Aided Engineering, Feng Chia University, Taichung 40724, Taiwan; cayuan@fcu.edu.tw; 2Department of Power Mechanical Engineering, National Tsing Hua University, Hsinchu City 30013, Taiwan; 0967356474shq@gmail.com; 3College of Semiconductor Research, National Tsing Hua University, Hsinchu City 30013, Taiwan

**Keywords:** Equation-Informed Neural Networks, advanced electronic packaging, numerical Bayesian Inference, constitutive equations, Pb-free SAC305 solders

## Abstract

Equation-Informed Neural Networks (EINNs) are developed as an efficient method for extracting the coefficients of constitutive equations. Subsequently, numerical Bayesian Inference (BI) iterations were applied to estimate the distribution of these coefficients, thereby further refining them. We could generate coefficients optimally aligned with the targeted application scenario by carefully adjusting pre-processing mapping parameters and identifying dataset preferences. Leveraging graphical representation techniques, the EINNs formulation is implemented in temperature- and strain-rate-dependent hyperbolic Garofalo, Anand, and Chaboche constitutive models to extract the corresponding coefficients for lead-free SAC305 solder material. The performance of the EINNs-based extracted coefficients, obtained from experimental results of SAC305 solder material, is comparable to existing studies. The methodology offers the dual advantage of providing the coefficients’ value and distribution against the training dataset.

## 1. Introduction

Proper material constitutive models and related coefficients are fundamental for reliable finite element predictions, encompassing the performance prediction model [1], the manufacturing process [2,3], and the reliability prediction models. Non-linear material properties, based on the temperature- and strain-rate-dependent material models, are often necessary for modeling critical sections of electronic packaging [4,5] and further influence the accuracy and predictability of the surrogate AI models [6,7,8].

Solder, a key component in electronic packaging, is often associated with potential fatigue failures. Wilde et al. conducted a study on the rate-dependent constitutive relationship of Pb-rich material [9], resulting in extracting Anand-based coefficients and identifying kinematic hardening, also known as the Bauschinger effect. To gain a better understanding of the creep characteristics of Pb-free solders, Xiao and Armstrong [10] performed tensile tests on both eutectic PbSn and Sn3.9Ag0.6Cu solder. Their findings revealed substantial microstructural alterations in the Sn3.9Ag0.6Cu with significantly lower absolute creep rates than the PbSn eutectic. The creep measurement data were successfully fitted into the Garofalo model [11], and the corresponding Garofalo coefficient was extracted.

Furthermore, Motalab et al. [12,13] conducted creep tests under meticulous control of the microstructure of the SAC305 solder without an oxidized surface, yielding a set of nine parameters for the Anand model. Basit et al. [14] utilized the Anand constitutive model with the extracted coefficients for solder joint lifetime prediction. The Chaboche material model [15], which considers the Bauschinger effect, was applied by Xie and Chen [16], Deshpande et al. [17], Wang et al. [5], and Yan et al. [18] for life prediction using Manson–Coffin type equations.

Ma and Suhling reviewed the constitutive equation and the corresponding coefficients of lead-free solder joint [19], and significant coefficient discrepancies have been reported. On the other hand, finite element engineers frequently face difficulties in selecting an appropriate material model and its parameters, as the measurement conditions may differ from those in practical applications. Kuczynska et al. [20] performed mechanical/dynamic tests against the solder joint to verify the ability of these material models and their coefficients to map the lifetime differences depending on the temperature rate under field and testing conditions, as well as on the mean operating temperature.

Considering the many application scenarios, which may range from low to high temperatures and strain rates, an emerging trend encourages users to obtain their own material coefficients [5]. This approach emphasizes the importance of tailoring the coefficients to the specific conditions encountered in each unique application. The least squares method and its derivatives are frequently employed in extracting coefficients. Although this approach is well established, matrix multiplication and inversion may diminish computational efficiency when handling extensive datasets. Moreover, the obtained coefficients based on the least square method are sensitive to the outliers, and this method does not apply to the censored data.

Historically, machine learning has harbored a certain resistance to rule-based inference. However, the efficacy of neural networks in symbolic computation is gaining recognition by integrating symbolic reasoning with continuous representations. Pioneers such as Zaremba et al. [21] and Allamanis [22] have explored the application of neural networks in handling mathematical objects. A significant advancement came from Lample and Charton [23] when they proposed a representation for mathematical expressions. Sharma et al., Chhabra et al., and Yadav et al. have applied neural network method for material optimization studies [24,25,26]. As a result, the theoretical basis for representing mathematical and symbolic equations using neural networks is well-established. 

In this research, we have developed the Equation-Informed Neural Networks (EINNs) method, synergistically incorporating the Bayesian Inference (BI) iteration technique to extract the coefficients of constitutive equations from measurement data. As visualized in Figure 1, the foundational concept of EINNs begins with constructing an artificial neural network to embody the constitutive equation f, where coefficient $\alpha_{k}$ is designated as their respective weights. Subsequently, EINNs deploy a pre-processing mapping technique on the input/output data pairs which are obtained from the experiments, a strategy rooted in neural network learning theory, and enables exploring coefficient fitting across various domains in-depth.

This neural network can be incrementally trained using input and output data pairs, facilitating the simultaneous approximation of coefficient $\alpha_{k}$. Theoretically, the steepest descent algorithm of the neural network backpropagation bolsters the computation efficiency and fosters the selective learning of data pairs. The final coefficients are obtained by the post-processing conversion. Utilizing the coefficients obtained by EINNs as initial values, Bayesian Inference (BI) is applied to obtain the distribution of the coefficients against the training datasets and further enhance the accuracy of coefficient extraction.

This paper is organized as follows: the “Theory” section provides an introduction to the framework of Equation-Informed Neural Networks (EINNs) and the numerical Bayesian Inference (BI) method. The subsequent section, “EINN Formulation”, presents the conversion process of constitutive equations from their conventional mathematical forms to their EINN equivalents, complete with pre-processing mapping and post-processing functions. In the “Applications” section, we apply the EINN formulation to the coefficient extraction of the material constitutive equation pertinent to Pb-free SAC305 solder joints. Detailed discussions and numerical results pertaining to the EINN formulations of the Chaboche, hyperbolic Garofalo, and Anand material models are also included. The paper concludes with a concise summary of our findings.

## 2. Theory

### 2.1. The Framework of Equation-Informed Neural Networks (EINNs)

Assume a constitutive equation is given by the function:(1)$$y_{i}=f(x_{j};\alpha_{k}),$$
where $y_{i}$, $x_{j}$, and $\alpha_{k}$ are vectors in real space with dimensions i, j, and k, respectively. The $x_{j}$ and $y_{i}$ represent the input and output of the functions, while $\alpha_{k}$ refers to the coefficients. Design pre-processing mapping functions:(2)$$M_{x}\left(x_{j}\right)=X_{j}\;\mathrm{and}\;M_{y}\left(y_{i}\right)=Y_{i},$$
which serve to effectively modify the domains of $x_{j}$ and $y_{i}$ to optimize the precision of coefficient extraction. Consequently, a new function can be formulated as $Y_{i}=F(X_{j},A_{k})$. Meanwhile, the corresponding neural network representations of $Y_{i}$ are formulated, and the coefficient $A_{k}$ is assigned as the weighting.

The learning process of the neural network involves continuous adjustment of these weights or coefficients. These adjustments can be computed for each known data pair using steepest-descent-based backpropagation as $A_{k}^{new}=A_{k}^{new}-\eta \Delta_{y}\frac{\partial y_{i}}{\partial A_{k}}$. Since these updates are independent of each data pair, the computationally expensive matrix multiplication and inversion inherent in the least squares-based approaches can be avoided. Furthermore, incorporating ratios into the adjustments allows for user emphasis on specific data pairs. This can be implemented as $A_{k}^{new}=A_{k}^{new}-\eta \;\sum_{l}r_{l}\cdot {\left(\Delta_{y}\frac{\partial y_{i}}{\partial A_{k}}\right)}^{\left(l\right)}\;$, where *l* is the coefficient adjustment from each data pair and $\sum_{l}r_{l}=1$.

Following several learning iterations with satisfactory accuracy, the coefficient $A_{k}$ of the constitutive equation can be obtained. However, due to the pre-processing mapping function (2) being applied, counteractions are required to reverse its effect. Therefore, we define the post-processing conversion functions as follows.
(3)$$a_{k}=g_{k}\left(x_{j},y_{i},A_{k}\right).$$

Through the combined application of pre-processing mapping functions and post-processing conversion of coefficients, the EINN framework gains an additional degree of freedom, bolstering the accuracy of coefficient extraction. Additionally, the steepest descent method offers a unique opportunity to prioritize specific data pairs while maintaining high computational efficiency.

### 2.2. The Numerical Bayesian Inference (BI) Iteration

We define the mean square error (MSE) function of Equation (1) with respect to the coefficients $\alpha_{k}$, as
(4)$$\epsilon \left(\alpha_{k}\right)=\sum_{l}{\left\|y_{t,i}^{\left(l\right)}-y_{i}\left(x_{j}^{\left(l\right)};\alpha_{k}\right)\right\|}^{2},$$
where $x_{j}^{l}$ and $y_{t,i}^{l}$ denote the input and ground truth of the l-th datapair, respectively. 

Assume that the distribution of the data pairs $y_{t,i}^{\left(l\right)}$ and $x_{j}^{\left(l\right)}$ are normal, and so is the error function $\epsilon \left(\alpha_{k}\right)$, denoted as $\epsilon \left(\alpha_{k}\right)\sim N(\mu ,\tau )$. Because the parameter τ cannot be negative, we assume it follows the gamma distribution, so that $\tau \sim G(a_{0},b_{0})$, where $a_{0}$ and $b_{0}$ are the gamma distribution parameters of τ. Moreover, assume that all the coefficients follow the normal distribution, say $\alpha_{k}\sim N\left(\mu_{k},\tau_{k}\right)$, and $\mu_{k}$ and $\tau_{k}$ are the average and precision, respectively. The posterior distribution after the BI remains normal distribution. In practice, we set $\mu_{k}$ equal to $\alpha_{k}$.

Consequently, the probabilities of the coefficient τ and $\alpha_{k}$ can be derived as $P\left(\tau \right)=\frac{b_{0}^{a_{0}}\tau^{a_{0}-1}e^{-b_{0}\tau }}{\Gamma \left(a_{0}\right)}$ and $P\left(\alpha_{k}\right)={\left(2\pi \right)}^{-\frac{1}{2}}\tau_{k}^{\frac{1}{2}}e^{-\frac{1}{2}\tau_{k}{\left(\alpha_{k}-\mu_{k}\right)}^{2}}$, respectively. The likelihood with respect to coefficient τ and $\alpha_{k}$ is $L\left(\alpha_{k},\tau \right)\equiv P\left(data|\alpha_{k},\tau \right)=\prod_{l=1}^{L}\frac{1}{\sqrt{2\pi }}\tau^{\frac{n}{2}}e^{-\frac{{\left(y_{t,i}^{\left(l\right)}-y_{i}^{\left(l\right)}\right)}^{2}}{2\sigma^{2}}}={\left(2\pi \right)}^{-\frac{n}{2}}\tau^{\frac{n}{2}}e^{-\frac{\tau }{2}\epsilon \left(\alpha_{k}\right)}$ [27]. 

The posterior of the τ distribution can be updated by the gamma–gamma conjugate: (5)$$a_{0}^{new}=a_{0}^{old}+\frac{n}{2}\;\mathrm{and}\;b_{0}^{new}=b_{0}^{old}+\frac{1}{2}\epsilon ,$$

As Equation (1) is not always a linear function, the posterior of coefficient $\alpha_{k}$ cannot always be computed by conjugate. Therefore, under the assumption that the value of $\Delta \alpha_{k}$ is relatively small, a numerical integration approach is applied:(6)$$\int_{0}^{\infty }L(\alpha_{k})\cdot P\left(\alpha_{k}\right)d\alpha_{k}\;\sim \sum_{n=1}^{n=N}L(\alpha_{k}^{\left(0\right)}+n\cdot \Delta \alpha_{k})\cdot P\left(\alpha_{k}^{\left(0\right)}+n\cdot \Delta \alpha_{k}\right)\cdot \Delta \alpha_{k},$$
where $\alpha_{k}^{\left(0\right)}$ is the minimal value of $\alpha_{k}$ and n is the number of the equal split between the assigned maximum and minimum $\alpha_{k}$ with a total of N splits. The posterior can then be obtained using normal distribution approximation.

We employ the Markov Chain Monte Carlo (MCMC) method to compute large hierarchical models requiring integration over many parameters. By applying the Gibbs sampling, the τ distribution parameters $a_{0}$ and $b_{0}$ are first updated through the conjugate (Equation (5)), and a new τ value will be sampled from the gamma distribution. Each $\alpha_{k}$ will be updated sequenently, and the new value will be accepted. Following thousands of iterations, every $\alpha_{k}$ exhibits a normal distribution. The mean value of this distribution is computed and assigned as the updated value for $\alpha_{k}$.

## 3. EINN Formulation

This section outlines the development of Equation-Informed Neural Network (EINN) formulations for the hyperbolic Garofalo, nine-parameter Anand, and Chaboche models, including pre-processing mapping and post-processing coefficient functions. 

### 3.1. Hyperbolic Garofalo Model

The conventional hyperbolic Garofalo constitutive equation can be written as:(7)$$\dot{\varepsilon_{p}}=C_{1}\cdot {\left[\sinh\left(C_{2}\sigma \right)\right]}^{C_{3}}\cdot e^{-\frac{Q}{RT}},$$
where $\dot{\varepsilon_{p}}$, σ, Q, R, and T represent the plastic strain rate, stress, activation energy, gas constant, and temperature, respectively. $C_{1}$, $C_{2}$, and $C_{3}$ are the coefficients that need to be extracted from the experimental data. 

We introduce $e=\dot{\varepsilon_{p}}\cdot e^{\frac{Q}{RT}}$ and accumulate the data pairs of {e} and {σ} from the experimental results. In order to proportional convert the original data to the [a,b+a] domain, the pre-processing matching functions are defined as follows:(8)$$M_{x}\left(\sigma \right)=\frac{\sigma -\sigma_{m}}{\Delta \sigma }b+a=x\;\mathrm{and}\;M_{y}\left(e\right)=\frac{e-e_{m}}{\Delta e}b+a=y,$$
where $\sigma_{m}$ and Δσ represent the minimal and maximum different values of set {σ}, and e and Δe correspond to set {e}. Parameters a and b are parts of pre-processing mapping, and a=0.001 and b=1 are assigned for this case. The values after the pre-processing are defined as x and y, respectively. Subsequently, a new function can be derived as:(9)$$y=C{\left[\sinh Ax\right]}^{n},$$

The corresponding neural network can be defined in Figure 2. The definition of the neurons is given in Table 1. 

Accordingly, the post-processing conversion of the coefficients can be approximated as $C_{1}=C\Delta e\cdot r_{2}^{n}$, $C_{2}=A\cdot b/\Delta \sigma$, and $C_{3}=n$, where $r_{2}=\left(\frac{Ab}{\Delta \sigma }-Aa\right)$. 

### 3.2. Anand Model

Anand et al. [28] proposed a set of viscoplastic constitutive equations for the rate-dependent deformation of metals. Recently, the Anand model has been extensively applied to microelectronic solders exhibiting large viscoplastic deformations. In addition to the activation energy, there are eight coefficients in the Anand model. A two-step approach is commonly employed to extract these eight coefficients [9,12,13].

The governing equation for the first step of the Anand model, including the ultimate tensile stress ($\sigma^{*}$), plastic strain rate (${\dot{\varepsilon }}_{p}$), activation energy (Q), and temperature (T), is expressed in Equation (10). $\hat{s}$, ξ, A, n, and m are the coefficients that need to be extracted.
(10)$$\sigma^{*}=\frac{\hat{s}}{\xi }{\left(\frac{{\dot{\varepsilon }}_{p}}{A}\cdot e^{\frac{Q}{RT}}\right)}^{n}\sinh^{-1}\left[{\left(\frac{{\dot{\varepsilon }}_{p}}{A}e^{\frac{Q}{RT}}\right)}^{m}\right],$$

Utilizing the same method as in the previous section, we assume $e_{0}={\dot{\varepsilon }}_{p}\cdot e^{\frac{Q}{RT}}$. Since the value of the strain rate is relatively small compared to other input parameters, a scaling factor R is applied, such that $e=\frac{e_{0}}{R}$. For consistency within this paper, the same activation function as in the previous section is assumed. The data pair of {e} and $\left\{\sigma^{*}\right\}$ is collected from the Motalab et al. [12,13]. An additional y and x are introduced to represent the output and input parameters, and the pre-processing mapping functions are defined as
(11)$$x=\frac{e-e_{m}}{\Delta e}b_{e}+a_{e}\;\mathrm{and}\;y=\frac{\sigma^{*}-\sigma_{m}^{*}}{\Delta \sigma^{*}}b_{\sigma }+a_{\sigma },$$
where $e_{m}$ and Δe are the minimal and maximum difference among set {e}, and so are $\sigma_{m}^{*}$ and $\Delta \sigma^{*}$ in $\left\{\sigma^{*}\right\}$. $a_{e}$, $b_{e}$, $a_{\sigma }$, and $b_{\sigma }$ are the mapping coefficients. By defining $\beta =\frac{\hat{s}}{\xi }$, the new function can be written as
(12)$$y=\beta^{*}{\left(\frac{x}{A^{*}}\right)}^{n^{*}}\sinh^{-1}\left[{\left(\frac{x}{A^{*}}\right)}^{m^{*}}\right],$$

Based on Equation (12), the EINN representation can be formulated as Figure 3. This network’s definitions are listed in Table 2. 

By defining $r=\sigma_{m}^{*}-a_{\sigma }\frac{\Delta \sigma^{*}}{b_{\sigma }}$, the post-processing of the coefficients can be written as follows: (13)$$\frac{1}{A}=\frac{1}{A^{*}}\left(\frac{bb_{e}}{\Delta e}+\frac{a_{e}}{avg\left(y\right)}\right)\cdot \frac{1}{R},\;n=n^{*},\;m=m^{*}\;\mathrm{and}\;\beta =\left[\frac{\Delta \sigma }{b_{\sigma }}\cdot \beta^{*}+\frac{r}{\beta^{*}\cdot avg\left(x\right)}\right]=\frac{\hat{s}}{\xi },$$
where avg(x) and avg(y) are the averges of {e} and $\left\{\sigma^{*}\right\}$. 

The governing equation of the second step of the Anand model is listed in (14), and $s_{0}$, a, and $h_{0}$ are the three remaining coefficients. The parameter c is defined in (15), and ξ is defined as the smallest positive real number to keep c<1.
(14)$$\sigma =\sigma^{*}-{\left[{\left(\sigma^{*}-cs_{0}\right)}^{1-a}+\left(a-1\right)\left\{\left(ch_{0}\right){\left(\sigma^{*}\right)}^{-a}\right\}\varepsilon_{p}\right]}^{1/(1-a)},$$
(15)$$c=\frac{1}{\xi }\sinh^{-1}{\left(\frac{\dot{\varepsilon_{p}}}{A}e^{\frac{Q}{RT}}\right)}^{m}\;,$$

We assume that $x=\sinh^{-1}\left[{\left(\frac{\dot{\varepsilon }}{A}e^{\frac{Q}{RT}}\right)}^{m}\right]$, $y=(\sigma^{*}-\sigma )$, $l=\sigma^{*}$, and $z=\varepsilon_{p}$, and the pre-processing mapping functions are defined as
(16)$$y=\frac{y-y_{m}}{\Delta y}b_{y}+a_{y},\;l=\frac{l-l_{m}}{\Delta l}b_{l}+a_{l},\;x=\frac{x-x_{m}}{\Delta x}b_{x}+a_{x}\mathrm{and}\;z=\frac{z-z_{m}}{\Delta z}b_{z}+a_{z},$$

By assuming 1−a=a′, the new function can be written as
(17)$$y={\left[{\left(\bar{l}+{\left(-\frac{s_{0}}{\xi }\right)}^{*}\cdot x\right)}^{a^{\prime *}}-a^{\prime *}\left\{\left({\left(\frac{h_{0}}{\xi }\right)}^{*}\cdot \bar{y}\right){\left(l\right)}^{a^{\prime *}-1}\right\}\cdot \bar{z}\right]}^{1/a^{\prime *}},$$

Based on Equation (17), the EINN representation can be formulated as Figure 4. This network’s definitions are listed in Table 3. Moreover, the post-processing of coefficients can be derived as
(18)$$a^{\prime *}=a\prime ,\;s_{0}^{\prime }=\frac{r_{y}}{r_{l}}\cdot (s_{0}),\;h_{0}^{\prime }=\frac{r_{y}}{r_{l}}\cdot r_{z}(h_{0}),$$
where $r_{l}=\frac{b_{l}}{\Delta l}$, $r_{x}=\frac{b_{x}}{\Delta x},r_{y}=\frac{b_{y}}{\Delta y}$, and $r_{z}=\frac{b_{z}}{\Delta z}$. 

### 3.3. Chaboche Model

The Chaboche model [15,29] is often applied for presenting the metallic material with the Bauschinger effect under cyclic loading. The original function can be written as
(19)$$\alpha =\frac{C}{\gamma }\left(1-e^{-\gamma {\cdot \varepsilon }_{p}}\right)+\sigma_{0}$$
where α and $\varepsilon_{p}$ are the back tensile stress and the plastic strain. $\sigma_{0}$ is the initial yielding stress, and C and γ are the fitting coefficients. To simplify the equation, we substitute and C/γ as β. Let $x=\varepsilon_{p}$, y=α, as the parameters, with the pre-processing mapping functions: (20)$$x=\frac{\varepsilon_{p}-\varepsilon_{p,m}}{\Delta \varepsilon_{p}}\;\mathrm{and}\;y=\frac{\alpha -\alpha_{m}}{\Delta \alpha },$$
where $\varepsilon_{p,m}$ and $\Delta \varepsilon_{p}$ are the minimal and maximum differences among set {$\varepsilon_{p}$}, and so are $\alpha_{m}$ and Δα in {α}, and $s=\sigma_{0}$. Hence, the new function can be re-written as
(21)$$y=\beta^{*}\left(1-e^{-\gamma^{*}\cdot x}\right)+s,$$
with the EINN formulation shown in Figure 5 and the neuron definition listed in Table 4.

Furthermore, the post-processing of coefficients can be derived as
(22)$$\sigma_{0}=x_{m}+s^{*}\cdot \Delta \alpha ,\;\gamma =\frac{\gamma^{*}}{\Delta \varepsilon_{p}},\;\mathrm{and}\;C=\frac{\Delta \alpha }{\Delta \varepsilon_{p}}\cdot (\beta^{*}\cdot \gamma^{*})$$

## 4. Applications

Building on the EINN formulation and Bayesian Inference (BI) iteration described in the preceding section, this chapter discusses the extraction of coefficients from the hyperbolic Garofalo, nine-parameter Anand, and Chaboche models for the SAC305 solder material. 

### 4.1. Hyperbolic Garofalo Model

The experimental dataset is drawn from Xiao and Armstrong [10]. To determine the coefficient C in Equation (9), we employ a grid search combined with a bisection optimization technique, whereas the EINN structure for coefficients A and n is addressed using standard backpropagation. To emphasize coefficient extraction for low temperatures (both 318 and 353 K) and low strain rates, ratios are assigned to the datapairs, as shown in Table 5. Table 5 also lists the input (plastic strain) and output (stress) of the EINN learning. Utilizing the post-processing conversion formula, the EINN coefficients, $C_{1}$, $C_{2}$, and $C_{3}$, are obtained and presented in the middle column of Table 6. The hyperbolic model, when compared to the experimental data, is depicted in Figure 6. The data at 388 K exhibits a more significant difference than the others, primarily due to the ratio setting outlined in Table 5.

A total of 1000 Bayesian Inference interactions were performed to obtain the distribution of the extracted coefficients. The distributions are displayed in Figure 7, represented as the ratio of each coefficient value to the average, and are expanded by the precision τ of the error function. As denoted by the dashed lines in Figure 7, which signify a 5% difference, stable distributions of coefficients $C_{2}$ and $C_{3}$ are observed, while the large variation in $C_{1}$ is attributed to the ratio setting, which induces a higher discrepancy among the 388 K data.

The coefficient extraction of the hyperbolic Garofalo constitutive equation highlights the flexibility of the EINN framework, as it allows for assigning ratios to data pairs to prioritize specific data. The fitting accuracy of the EINN results demonstrates a significant improvement compared to the original reports [10] as indicated by the mean square error (MSE) of Table 6, followed by Equation (4). Although the distribution of the $C_{1}$ coefficients demonstrate a small fraction of the outliers from BI integration as Figure 7, both $C_{2}$ and $C_{3}$ show statistical difference within ±5% difference. Over 1000 iterations, only 58 instances of $C_{1}$ shows more than ±5% difference of the average value. Consequently, a robust set of coefficients for the hyperbolic Garofaolo constitutive model is achieved. 

### 4.2. Anand Model

In this section, the Anand constitutive model coefficients extraction is implemented for the lead-free SAC305 solder. The same activation energy as in the previous section is applied for the sake of research consistency. To extract the remaining eight coefficients of the Anand constitutive model, the first step involves utilizing temperature and strain rate-dependent ultimate tensile stresses to determine the initial four coefficients. Subsequently, the second step defines the remaining parameters based on temperature and strain rate-dependent stress and plastic strain.

The experimental data are sourced from Motalab et al. [12]. The EINN formulation, following Equation (12), is applied with the pre-processing mapping coefficients $a_{e}$, $b_{e}$, $a_{\sigma }$, and $b_{\sigma }$ (Equation (13)) which are 0.8, 0.15, 0.9, and 0.1. It is vital to note that the selection of these mapping coefficients depends on the numerical characteristics of the dataset, and it is essential for preventing numerical errors during the backpropagation-based machine learning of the EINN formulation. 

During the learning phase of the EINN formulation, the coefficients $n^{*}$, $m^{*}$, $A^{*}$, and β of Equation (12) and Figure 3 are constrained to be positive. A grid search technique is employed to identify optimal initial values concerning the experimental data.

Furthermore, learning ratios are implemented to emphasize the learning preference for low strain rates and temperatures close to the working temperature of electronic components. After hundreds of iterations, the EINN coefficients are reported in Table 7. The MSE values indicate that the coefficients obtained from the EINN formulation exhibit similar accuracy to those obtained using conventional methods. The obtained step 1 Anand model is plotted in Figure 8. 

The EINN formulation coefficients serve as initial inputs for Bayesian Inference (BI) to analyze the statistical distribution of the coefficients. Figure 9 illustrates the distribution of the coefficients, with dashed lines indicating differences within ±5%. Due to its low value, the coefficient n was not examined. Both coefficients A and β exhibit distribution within ±5% difference. Out of 1500 values, only 61 cases of coefficient m exceed ±5% difference, which can be attributed to the preference settings during the EINN learning process. The average coefficients obtained from BI are presented in the last column of Table 7 and are utilized for the subsequent coefficient extraction step in the Anand model.

The temperature and strain rate dependent stress–strain curves are obtained from Motalab [12]. The EINN formulation of the step 2 Anand model, as indicated in Equation (17) and Figure 4, is applied with the pre-processing mapping parameters shown in Table 8, based on Equation (16), while in the EINN learning procedure, the values of $s_{0}$ and $h_{0}$ are forced to be positive. A grid search technique is applied to define the optimal initial coefficients. The learning ratios are implemented to emphasize the learning preference for low strain rates and temperatures close to the working temperature of electronic components, following the coefficient extraction strategy of Motalab et al. [12]. With Equation (18), the optimized coefficients can be obtained, as listed in Table 9, and the stress–strain curves at different strain rates from the Anand model are plotted against the experiment [12], as shown in Figure 10.

The dataset with high preference is applied to the BI iteration to mitigate the large coefficient shifting. Figure 11 plots the MSEs of EINNs and EINNs with BI against the Anand coefficient obtained by Motalab et al. [12], under different temperatures and strain rates. By adjusting the ratio of EINN network learning, the coefficient extraction can be fine-tuned to perform better in the room to the working temperature at a low strain rate, as indicated in Figure 11. 

### 4.3. Chaboche Model

To study the lifetime of the ball-grid-array-type of advanced electronic packaging, the Chaboche material model is often applied [5,8]. The Chaboche model and its coefficients can be extracted from the temperature-dependent stress–strain curves by a given strain rate. Unlike the previous sectors, this section investigates the extraction of Chaboche coefficients from the Anand model.

The Anand coefficients from Table 6 and Table 7, adjusted via Bayesian Inference (BI), are utilized to generate inputs for the Chaboche model. A strain rate of $10^{-5}$ (1/s) is maintained, given that the Anand coefficients have been optimized for lower strain rates, as demonstrated in the previous section. Stress–strain curves can be generated by the Anand model (as Equations (14) and (15)) for each temperature point, including −40 °C, −20 °C, 40 °C, 80 °C, and 122 °C.

The temperature-dependent stress–strain data serve as the training datasets. With the pre-processing mapping established by Equation (20), we apply the EINN formulation for the Chaboche model as Equation (21). Following this, the steepest-descent coefficient optimization is applied to the EINN formulation (as illustrated in Figure 5) with the neural definitions outlined in Table 4. The post-processing of the coefficients Equation (22) allows for the acquisition of Chaboche coefficients at various temperatures. The resultant data are documented in Table 10, with the mean square errors (MSE) compared to the input dataset.

The coefficients derived from the EINN formulation are subsequently incorporated into Bayesian Inference (BI) iterations for the temperature-dependent Chaboche model. Figure 12 delineates the distribution of coefficients $\sigma_{0}$, C, and γ across different temperatures, magnified by the precision τ of the error function. The vertical axes in this figure represent the ratio of the coefficient value obtained at each BI iteration to the averaged value. Table 11 contains the averaged coefficient post-BI.

While variations in all coefficients lie within a ±5% difference, a larger variety, coupled with a lower MSE, as listed in Table 10 and Table 11, is evident at higher temperatures. This suggests a reduced coefficient sensitivity at these elevated temperatures. By introducing Young’s modulus obtained by linear extrapolation from the experiment [12], the temperature-dependent stress–strain curves are plotted in Figure 13. 

## 5. Conclusions

In this study, we developed the concept of Equation-Informed Neural Networks (EINNs) as an efficient method for extracting the coefficients of constitutive equations. Subsequently, the MCMC with numerical Bayesian Inference (BI) iterations was applied to estimate the distribution of these coefficients, thereby further refining them.

The EINN formulation was derived by leveraging graphical representation techniques to convert the mathematical form of constitutive equations into an equivalent EINN format. By carefully adjusting pre-processing mapping parameters and identifying dataset preferences, we could generate coefficients optimally aligned with the targeted application scenario.

The EINN formulation has been successfully applied to the hyperbolic Garofalo, Anand, and Chaboche constitutive models. This paper details the EINN formulation with its neural network format, the definition of each neuron, the appropriate pre-processing techniques, and the post-processing of the coefficients.

The extraction of coefficients for the hyperbolic Garofalo and Anand models was conducted using experimental results from lead-free SAC305 solder material studies by Xiao and Armstrong [10] and Motalab et al. [12,13]. Our report includes the employed pre-processing mapping techniques and parameters. With the dataset preference, the constitutive equations with extracted coefficients performed better in the interested zone.

Comparisons with coefficients of the constitutive equations from the aforementioned studies demonstrated that those extracted from the EINN formulation were alike. Importantly, the mean square error (MSE) of the EINN formulation learning was comparable to those from the literature [10,12,13]. The performance of the MSE depends on many factors, such as the prescription capability of the material model and experimental measurement accuracy. In this research, the MES is applied as a comparison of how the coefficients extracted by the EINNs perform to the ones obtained by the original methods.

Moreover, the MCMC with numerical Bayesian Inference (BI) iteration technique was employed to analyze the robustness of the extracted coefficients against the experiment data, as shown in Figure 7, Figure 9 and Figure 12. A slightly higher variation was observed when the dataset preference was applied to the EINN learning. Nevertheless, the coefficients derived from EINNs remained within a ±5% confidence interval.

In conclusion, the combined use of EINNs with BI provides a powerful tool for extracting coefficients from temperature- and strain-rate-dependent constitutive equations with dataset preference. This is under the assumption that the SAC305 solder material characteristics can be described by the material model and that the experimental measurement is accurate enough. This approach provides the coefficients’ value and the distribution of coefficients against the training dataset.

This study’s potential limitations may include the dataset preference assumption, which may not universally apply across all scenarios. Additionally, the applicability of the EINN formulation to all forms of constitutive equations remains to be fully determined, necessitating further exploration of potential limitations. Moreover, advanced neural network backpropagation methods, such as Levenberg–Marquardt (LM) algorithm, will be applied to EINN frameworks. 

## Figures and Tables

**Figure 1 materials-16-04922-f001:**
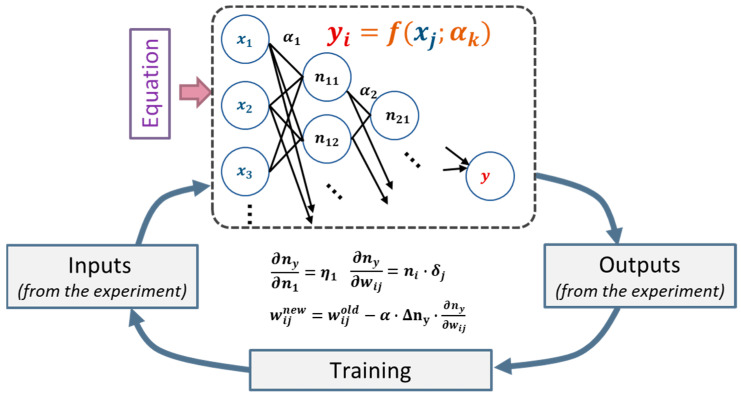
The concept of Equation-Informed Neural Networks (EINNs).

**Figure 2 materials-16-04922-f002:**
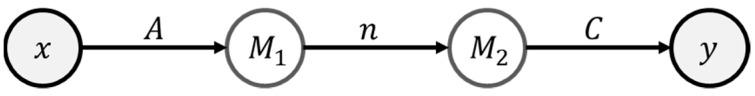
The EINNs for hyperbolic Garofalo model.

**Figure 3 materials-16-04922-f003:**
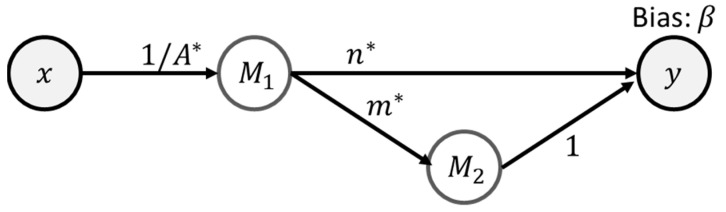
The EINNs for step 1 Anand equation.

**Figure 4 materials-16-04922-f004:**
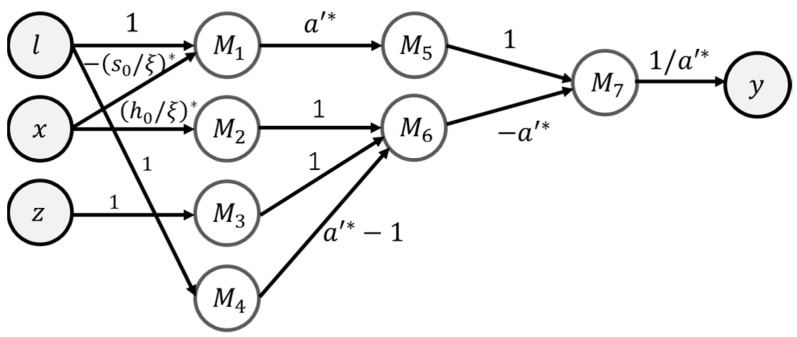
The EINNs for the step 2 Anand equation.

**Figure 5 materials-16-04922-f005:**
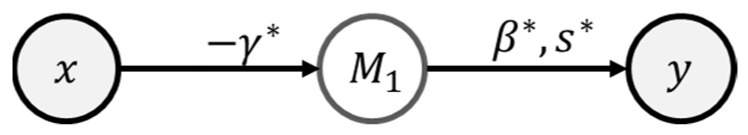
The EINN formulation for the Chaboche model.

**Figure 6 materials-16-04922-f006:**
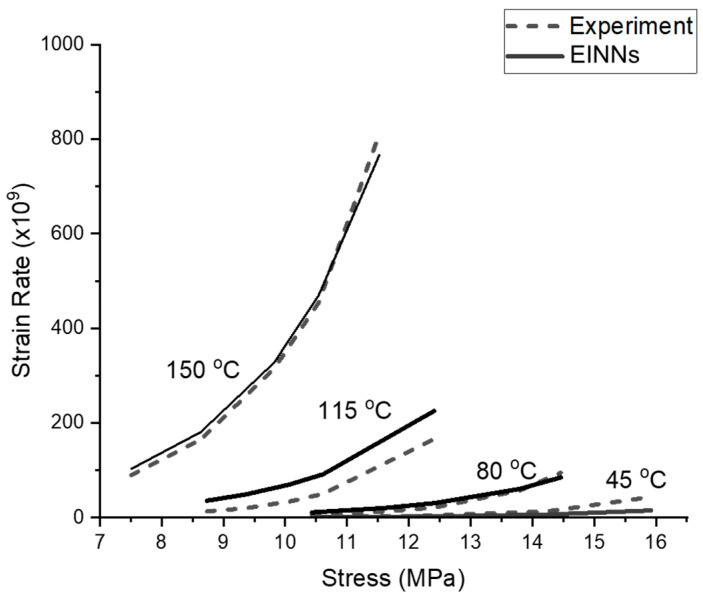
The obtained hyperbolic Garofalo curves for different temperatures. The experimental data are from Xiao and Armstrong [10].

**Figure 7 materials-16-04922-f007:**
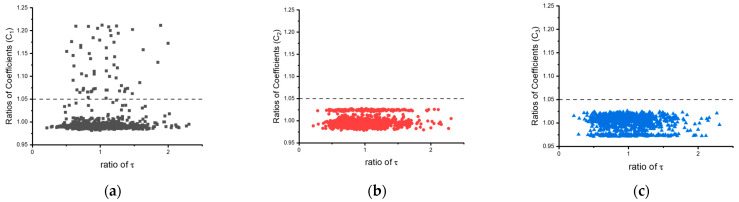
The distribution of the coefficients from the BI iteration. (**a**–**c**) are coefficients $C_{1}$, $C_{2}$, and $C_{3}$, respectively.

**Figure 8 materials-16-04922-f008:**
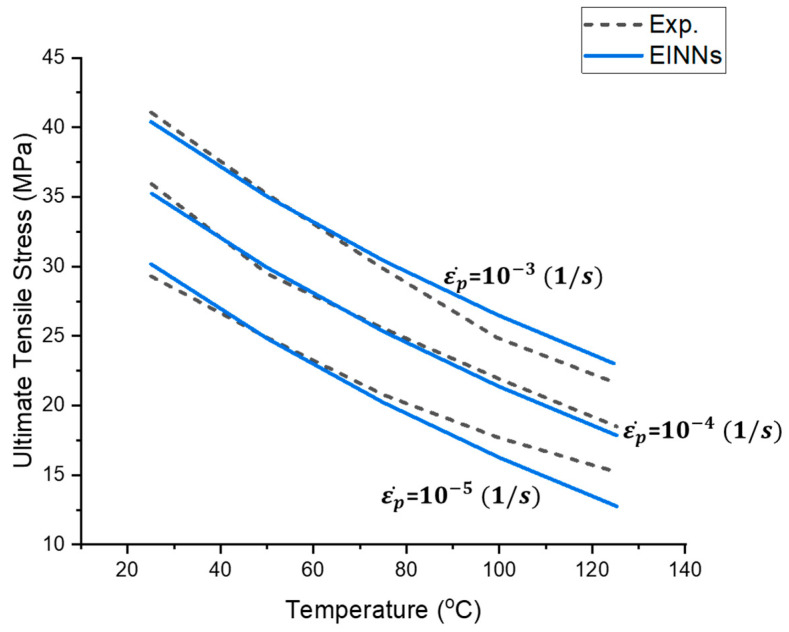
Step 1 Anand model at different temperatures. The experimental data are based on Motalab et al. [12].

**Figure 9 materials-16-04922-f009:**
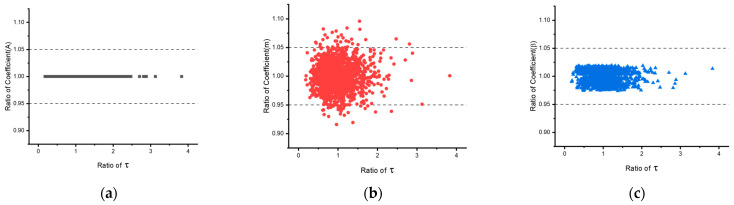
The distribution of the coefficients from the BI iteration. (**a**–**c**) are coefficients A, m, and β, respectively.

**Figure 10 materials-16-04922-f010:**
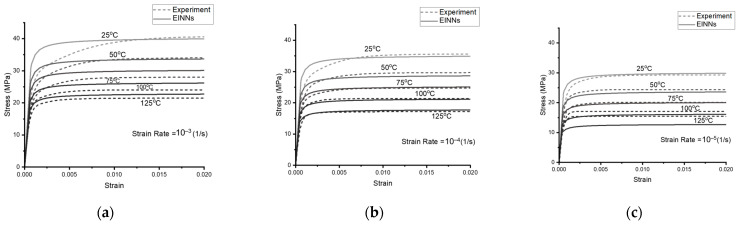
The obtained step 2 Anand model curves for different temperatures. (**a**–**c**) are the obtained Anand model with strain rates of $10^{-3}$, $10^{-4}$, and $10^{-5}$ (1/s). The experimental data are based on Motalab et al. [12].

**Figure 11 materials-16-04922-f011:**
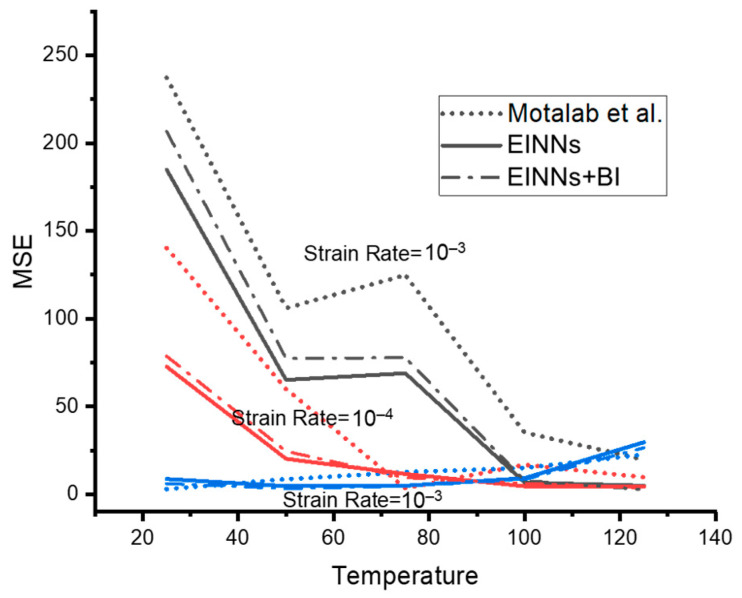
The MSE plot of step 2 Anand model coefficient extraction results Motalab et al. [12].

**Figure 12 materials-16-04922-f012:**
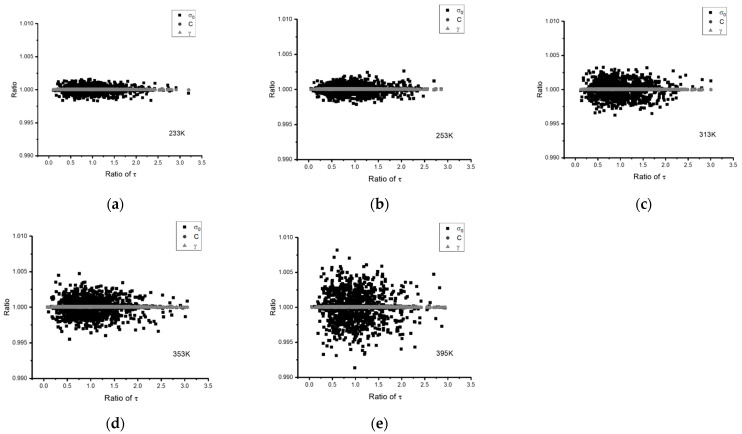
The distributions of the temperature-dependent Chaboche coefficients obtained by BI. (**a**–**e**) represent the distributions of −40, −20, 40, 80, and 122 °C, respectively.

**Figure 13 materials-16-04922-f013:**
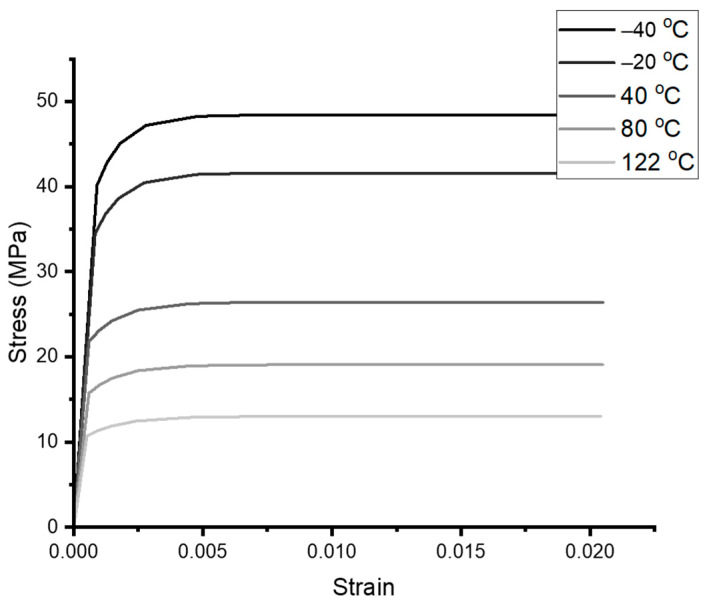
The temperature-dependent stress–stress curves from Chaboche model using the coefficients in Table 11.

**Table 1 materials-16-04922-t001:** The neuron definition of the EINN representation of the hyperbolic Garofalo equation.

Neuron	Net Value	Activation
$M_{1}$	$M_{1,net}=A\cdot x$	$M_{1}=ln(\sinh M_{1,net})$
$M_{2}$	$M_{2,net}=n\cdot M_{1}$	$M_{2}=e^{M_{2,net}}$
y	$y_{net}=C\cdot M_{2}$	$y=y_{net}$

**Table 2 materials-16-04922-t002:** The neuron definition of the EINN representation of the step 1 Anand equation.

Neuron	Net Value	Activation
$M_{1}$	$M_{1,net}={1/A}^{*}\cdot x$	$M_{1}=ln(M_{1,net})$
$M_{2}$	$M_{2,net}=m^{*}\cdot M_{1}$	$M_{2}=\ln\left(\sinh^{-1}e^{M_{2,net}}\right)$
y	$y_{net}=n^{*}\cdot M_{1}+M_{2}+\beta$	$y=e^{y_{net}}$

**Table 3 materials-16-04922-t003:** The neuron definition of the EINN representation of the step 2 Anand equation.

Neuron	Net Value	Activation
$M_{1}$	$M_{1,net}=l+{\left(-\frac{s_{0}}{\xi }\right)}^{*}x$	$M_{1}=ln(M_{1,net})$
$M_{2}$	$M_{2,net}={\left(\frac{h_{0}}{\xi }\right)}^{*}\cdot M_{1}$	$M_{2}=ln(M_{2,net})$
$M_{3}$	$M_{3,net}=$ z	$M_{3}=ln(M_{3,net})$
$M_{4}$	$M_{4,net}=l$	$M_{4}=ln(M_{4,net})$
$M_{5}$	$M_{5,net}=a^{\prime *}\cdot M_{1}$	$M_{5}=e^{M_{5,net}}$
$M_{6}$	$M_{6,net}=M_{2}+M_{3}+(a^{\prime *}-1)\cdot M_{4}$	$M_{6}=e^{M_{6,net}}$
$M_{7}$	$M_{7,net}=M_{5}-a^{\prime *}\cdot M_{6}$	$M_{7}=\ln\left(M_{7,net}\right)$
y	$y_{net}=\frac{1}{a^{\prime *}}\cdot M_{2}$	$y=e^{y_{net}}$

**Table 4 materials-16-04922-t004:** The neuron definition of the EINN representation of the Chaboche equation.

Neuron	Net Value	Activation
$M_{1}$	$M_{1,net}=(-\gamma^{*})\cdot x$	$M_{1}=1-e^{M_{1,net}}$
y	$y_{net}=\beta^{*}\cdot M_{1}+s^{*}$	$y=y_{net}$

**Table 5 materials-16-04922-t005:** The ratios applied to the data pair to emphasize the preference.

Data ID	Temperature (°C)	$\mathrm{Plastic}\;\mathrm{Strain}\;\mathrm{Rate}\;(\dot{\varepsilon_{p}},10^{-9}1/s)$	Stress (MPa)	Ratio
1	45	1.4	10.54	18.0
2	45	4.0	12.30	6.0
3	45	13.6	14.25	5.0
4	45	43.9	15.92	0.5
5	80	6.6	10.43	18.0
6	80	13.1	11.59	6.0
7	80	21.2	12.40	5.0
8	80	57.8	13.80	3.0
9	80	95.0	14.46	1.0
10	115	13.4	8.73	15.0
11	115	19.8	9.35	12.0
12	115	34.2	10.07	6.0
13	115	50.5	10.61	6.0
14	115	166.0	12.41	1.0
15	150	90.1	7.51	1.0
16	150	165.0	8.63	1.0
17	150	315.0	9.82	1.0
18	150	454	10.54	0.5
19	150	810	11.52	0.5

**Table 6 materials-16-04922-t006:** The comparison of the extracted coefficients of Hyperbolic Garofalo Model.

	Xiao and Armstrong [10]	EINNs	EINNs + BI
Q (kJ/mol)	62,000	65,000	65,000
$C_{1}$	0.184	0.539	0.443
$C_{2}$	0.221	0.473	0.482
$C_{3}$	2.89	1.055	1.073
*MSE* *	37,188.5	11,794.9	10,967.4

*: defined by Equation (4).

**Table 7 materials-16-04922-t007:** The comparison of the extracted coefficients of step 1 Anand model.

	Motalab et al. [12]	EINNs	EINNs + BI
A	3501	1650	1649
n	$1.00\times 10^{-2}$	$1.54\times 10^{-4}$	$1.64\times 10^{-4}$
m	0.25	0.54	0.53
$\beta =\hat{s}/\xi$	7.55	4.11	4.16
*MSE* *	17.03	15.96	15.78

*: defined by Equation (4).

**Table 8 materials-16-04922-t008:** The pre-processing mapping parameters.

	y	l	x	z
a	1	0.8	0.1	0.8
b	0	0.4	0.05	0.1

**Table 9 materials-16-04922-t009:** The comparison of the extracted coefficients of step 2 Anand model.

	Motalab et al. [12]	EINNs	EINNs + BI
ξ	4	17.66	17.66
$s_{0}$	21	55.96	57.45
a	1.78	2.30	2.26
$h_{0}$	18,000	828,822	828,822

**Table 10 materials-16-04922-t010:** Temperature-dependent Chaboche coefficients of EINNs.

Temperature	$\sigma_{0}$	C	γ	MSE *
−40 °C	39.32	9174.1	1004.7	1.11
−20 °C	33.80	7535.7	964.0	0.84
40 °C	21.35	4216.0	840.0	0.50
80 °C	15.43	2988.0	824.3	0.28
122 °C	10.50	1886.5	759.1	0.18

*: defined by Equation (4).

**Table 11 materials-16-04922-t011:** Temperature-dependent Chaboche coefficients of EINNs and BI.

Temperature	$\sigma_{0}$	C	γ	MSE *
−40 °C	39.30	9174.1	1004.7	1.11
−20 °C	33.78	7535.7	964.0	0.84
40 °C	21.39	4216.0	840.0	0.49
80 °C	15.45	2988.0	824.3	0.27
122 °C	10.53	1886.5	759.1	0.17

*: defined by Equation (4).

## Data Availability

The data presented in this research study are available in this article.

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
