# Peer review of "Coefficient Extraction of SAC305 Solder Constitutive Equations Using Equation-Informed Neural Networks"

_materials, 2023, doi:10.3390/ma16144922_

Round 1

Reviewer 1 Report

Authors are advised to double check the manuscript to avoid typing and grammatical errors

Reviewer 2 Report

In this manuscript, authors developed the concept of equation-informed neural networks (EINS) as an efficient method for extracting the coefficients of constitutive equations. Subsequently, numerical Bayesian inference (BI) iterations were applied to estimate the distribution of these coefficients, thereby further refining them. The idea seems interesting, however some points were not clear. Although a comparison with the literature was presented, the references used were published more than a decade ago. This practice leaves the scientific contribution of the manuscript in doubt. What is the state of the art in the subject addressed?

Some additional comments

1. According to the International System of Units (SI) 9th edition 2019,

a) When the symbol % is used, a space separates the number and the symbol %.

b) The numerical value always precedes the unit and a space is always used to separate the unit from the number. Thus, the value of the quantity is the product of the number and the unit. The space between the number and the unit is regarded as a multiplication sign (just as a space between units implies multiplication). The only exceptions to this rule are for the unit symbols for degree, minute and second for plane angle, °, ′ and ″, respectively, for which no space is left between the numerical value and the unit symbol.

These comments are valid for the whole manuscript.

c) Following the 9th CGPM (1948, Resolution 7) and the 22nd CGPM (2003, Resolution 10), for numbers with many digits the digits may be divided into groups of three by a space, in order to facilitate reading. Neither dots nor commas are inserted in the spaces between groups of three. However, when there are only four digits before or after the decimal marker, it is customary not to use a space to isolate a single digit. The practice of grouping digits in this way is a matter of choice; it is not always followed in certain specialized applications such as engineering drawings, financial statements and scripts to be read by a computer.

d) Although temperature can be expressed in K and °C, both units accepted by the SI, standardization is recommended.

2. Take care with the manuscript formatting. Tables 5 and 7 were divided into two pages. There are subtitles that were left at the end of the page.

Also check the writing.

3. Check the number of significant digits in the results shown in Tables 5 and 7.

4. Check the writing of the title of Figure 12. Lack of space between declared temperature values.

5. On page 2, Line 67 t0 67 “Moreover, the obtained coefficients based on the least square method are sensitive to the outliners, and this method does not apply to the censored data.”

Outliners or outliers? Check!

6. On Page 10, Line 271 “Although the distribution of the ??1 coefficients demonstrate a small faction of the outliners from BI integration as Figure 7, both ??2 272 and ??3 show statistical robustness within ±5% difference.”

a) Faction? Check?

b) Outliners? Check!

c) What does “statistical robustness” mean?

d) Difference or residuals?

7. The quality of Figure 8 should be improved.

8. What is the justification for using the experimental data from Motalab et al. [12,13]. These articles were published a decade ago. Xiao and Armstrong [10] was published in 2005. Other data from the literature could not have been used for this purpose as well? Several references published in the last 5 years are presented in the reference list.

a) Did you have access to numerical values of the variables of interest?

b) It is recommended to specify in each case which data were used.

c) Was the used dataset sufficient for validation?

Reviewer 3 Report

The paper addresses important issue of identification of constitutive equations using neural network. In my opinion the paper is interesting. Nevertheless, I have some suggestion:

Application of neural networks have recently gained a lot of interest in the context of material modelling. In my opinion authors should emphasise more novelty of  their paper  against the background of the  state of the art.  In my opinion introduction should be enriched by the state of the art on neural networks approaches in this application.

Could you clarify in the text what is the data structure of inputs and outputs? Are this sampling points of single experimental curve (what type of experiment)? Are 1 000 Bayesian interfence interactions performed on the same data or different samples? If different samples, have  you checked error of single identification ?

MSE in some case is very high. Could you comment more on that in the paper?

Round 2

Reviewer 1 Report

The authors have corrected the manuscript according to comments. Now, the manuscript may be accepted in its present form.

Reviewer 2 Report

All my comments were satisfactorily addressed by the authors. I recommend the  manuscript publication.

Reviewer 3 Report

In my opinion the paper could be published in the current form.